High-level expression and molecular characterization of a recombinant prolidase from Escherichia coli NovaBlue

Wang Tzu-Fan 1
Chi Meng-Chun 1
Lai Kuan-Ling 1 2
Lin Min-Guan 3
Chen Yi-Yu 1
Lo Huei-Fen 2 hflo@sunrise.hk.edu.tw
Lin Long-Liu 1 llin@mail.ncyu.edu.tw
1 Department of Applied Chemistry, National Chiayi University , Chiayi , Taiwan
2 Department of Food Science and Technology, Hungkuang University , Taichung , Taiwan
3 Institute of Molecular Biology, Academia Sinica , Taipei , Taiwan
Uversky Vladimir
Electronic publication date: 2018 Oct 31
Publication date: 2018
Volume: 6
Electronic Location ID: e5863
Received 2018 Aug 22; Accepted 2018 Oct 3
Copyright: © 2018 Wang et al.
Copyright year: 2018
Copyright holder: Wang et al.
License: This is an open access article distributed under the terms of the Creative Commons Attribution License, which permits unrestricted use, distribution, reproduction and adaptation in any medium and for any purpose provided that it is properly attributed. For attribution, the original author(s), title, publication source (PeerJ) and either DOI or URL of the article must be cited.
License URL: https://creativecommons.org/licenses/by/4.0/

Keywords: Escherichia coli, Gene expression, Prolidase, Organic co-solvents, Chaotropic agent-induced denaturation, Molecular characterization

Funding: MOST 106-2815-C-241-037-B MOST 107-2313-B-415-013 This work was supported by a college student research scholarship (MOST 106-2815-C-241-037-B) and research grant (MOST 107-2313-B-415-013) from the Ministry of Science and Technology of Taiwan. The funders had no role in study design, data collection and analysis, decision to publish, or preparation of the manuscript.

==============================
Long-term use of organophosphorus (OP) compounds has become an increasing global problem and a major threat to sustainability and human health. Prolidase is a proline-specific metallopeptidase that can offer an efficient option for the degradation of OP compounds. In this study, a full-length gene from Escherichia coli NovaBlue encoding a prolidase (EcPepQ) was amplified and cloned into the commercially-available vector pQE-30 to yield pQE-EcPepQ. The overexpressed enzyme was purified from the cell-free extract of isopropyl thio-β-D-galactoside IPTG-induced E. coli M15 (pQE-EcPepQ) cells by nickel-chelate chromatography. The molecular mass of EcPepQ was determined to be about 57 kDa by 12% sodium dodecyl sulfate–polyacrylamide gel electrophoresis and the result of size-exclusion chromatography demonstrated that the enzyme was mainly present in 25 mM Tris–HCl buffer (pH 8.0) as a dimeric form. The optimal conditions for EcPepQ activity were 60 °C, pH 8.0, and 0.1 mM Mn2+ ion. Kinetic analysis with Ala-Pro as the substrate showed that the Km and kcat values of EcPepQ were 8.8 mM and 926.5 ± 2.0 s−1, respectively. The thermal unfolding of EcPepQ followed a two-state process with one well-defined unfolding transition of 64.2 °C. Analysis of guanidine hydrochloride (GdnHCl)-induced denaturation by tryptophan emission fluorescence spectroscopy revealed that the enzyme had a [GdnHCl]0.5,N-U value of 1.98 M. The purified enzyme also exhibited some degree of tolerance to various water/organic co-solvents. Isopropanol and tetrahydrofuran were very detrimental to the enzymatic activity of EcPepQ; however, other more hydrophilic co-solvents, such as formamide, methanol, and ethylene glycol, were better tolerated. Eventually, the non-negative influence of some co-solvents on both catalytic activity and structural stability of EcPepQ allows to adjust the reaction conditions more suitable for EcPepQ-catalyzed bioprocess.

Introduction

Prolidases (Xaa-Pro dipeptidases, EC 3.4.13.9) act to hydrolyze all dipeptides with a non-polar amino acid at the N-terminus and proline or hydroxyproline at the C-terminus (Lowther & Matthews, 2002). This type of enzyme is widespread in nature and has been isolated from different mammalian tissues (Sjöström, Noŕen & Josefsson, 1973; Browne & O’Cuinn, 1983; Endo et al., 1987; Lupi et al., 2006) as well as from a variety of microorganisms, including the species of Escherichia (Park et al., 2004), Xanthomonas (Suga et al., 1995), Lactobacillus (Fernandez-Espla, Martin-Hernandez & Fox, 1997; Huang & Tanaka, 2015), Pyrococcus (Ghosh et al., 1998; Theriot et al., 2010), and Aspergillus (Jalving et al., 2002). Although the exact physiological role of prokaryotic prolidases remains to be elucidated, eukaryotic enzymes are thought to be involved in the recycling of collagen-derived proline (Hu, Phang & Valle, 2008). In this regard, activity fluctuations in eukaryotic-type prolidase can serve as an indicator of dysfunctional collagen metabolism as well as disease progression (Kitchener & Grunden, 2012; Namiduru, 2016).

As one of the most important components in pesticides and chemical warfare agents, organophosphorus (OP) compounds are highly toxic toward mammalian species (Kumar et al., 2010) and usually difficult to degrade through the natural metabolic processes of microorganisms (Karpouzas & Singh, 2006). Two well-known examples of OP compounds are the acetylcholinesterase inhibitor, diisopropylfluorophosphate, and G-series nerve agents, such as sarin and soman (Ganesan, Raza & Vijayaraghavan, 2010). Several microbial enzymes, particularly organophosphorus acid anhydrolase (OPAA) and phosphotriesterase, have been biochemically characterized for their high activity and unique substrate specificity against G-series nerve agents (Singh, 2009; Kitchener & Grunden, 2012). A recent review has documented that the three-dimensional structures of organophosphate-degrading metallohydrolases have a common “pita-bread” fold with a dinuclear metal center coordinated by five highly conserved residues (Schenk et al., 2016). Interestingly, the crystal structures of microbial prolidases also share the same “pita-bread” fold with the characteristic catalytic dinuclear metal site (Maher et al., 2004; Jeyakanthan et al., 2009; Weaver et al., 2014; Kgosisejo et al., 2017). Examination of the crystal structure of the nerve agent degrading OPAA/prolidase from Alteromonas sp. strain JD6.5 further helps us to understand the structural conservation between OPAA and prolidase (Vyas et al., 2010). Over the last two decades, some important advances in our understanding of the biotechnological application of prolidases have already been exerted on their intrinsic ability to hydrolyze toxic OP-containing compounds (DeFrank & Cheng, 1991; Cheng et al., 1998; DiTargiani et al., 2010; Theriot et al., 2011; Chandrasekaran, Belinskaya & Saxena, 2013; Yuh et al., 2017). This fact renders prolidases very suitable to act as a catalytic bioscavenger for alleviating OP nerve agent toxicity (Theriot et al., 2010; DiTargiani et al., 2010; Theriot et al., 2011).

Previously, some catalytic properties of Escherichia coli BL21 (DE3) prolidase (EcPepQ) has been characterized (Park et al., 2004). The recombinant enzyme exhibits hydrolytic activity against a series of OP triesters so that it has been proposed to be useful for the kinetic resolution of racemic phosphate esters. Structural basis for the substrate selectivity of EcPepQ has also been elucidated just a few years ago (Weaver et al., 2014). Through docking simulations and site-directed mutagenesis, Weaver and his co-workers have suggested that the location of the loop R370 in EcPepQ plays an important part in the evolution of enzyme selectivity. Until now, to the best of our knowledge, only the aforementioned studies have investigated the molecular properties of E. coli prolidase. Given the fact that more information about the enzymatic characteristics of EcPepQ is essential to ensure a satisfactory application of this enzyme, a full-length gene encoding E. coli NovaBlue prolidase was amplified and cloned into the pQE-30 expression vector. The overexpression of EcPepQ was induced with isopropyl thio-β-D-galactoside (IPTG) and the recombinant enzyme was purified from the cell-free extract by affinity chromatography on nickel-nitrilotriacetate (Ni-NTA) resin. In addition, the biochemical and biophysical properties of EcPepQ were successively examined in order to explore its biotechnological potential.

Materials and Methods

Expression and purification of the recombinant enzyme

Chromosomal DNA from E. coli NovaBlue was isolated by a commercial kit (Geno Plus Genomic DNA Extraction Miniprep System; Viogene Inc., Valencia, CA, USA) and used as a template for the amplification of the pepQ gene by polymerase chain reaction (PCR). PCR amplification was performed on Applied Biosystems 2400 Thermal Cycler (Thermo Fisher Scientific Inc., Waltham, MA, USA) with two sets of primers (PepQ-F: 5′-GGATCCATGGAATCACTGGCCTCGCTC and PepQ-R: 5′-AAGCTTTCACGCCAGTTTCAGATCCCG), designed on the basis of the gene sequence (Accession number: P21165) deposited in GenBank. Touchdown PCR was conducted as follows: an initial denaturation at 94 °C for 3 min, followed by 30 cycles of denaturation of 94 °C for 2 min, annealing at 53 °C for 1.5 min, and extension at 72 °C for 2 min. After a final extension at 72 °C for 2 min, the amplified fragment was digested with BamHI and HindIII, and inserted into the predigested vector pQE-30 (Qiagen, Hilden, Germany) to yield pQE-EcPepQ.

Single colony of E. coli M15 (pQE-EcPepQ) was inoculated to Luria-Bertani (LB) medium supplemented with 100 μg/mL ampicillin and 25 μg/mL kanamycin and the culture was shaken at 37 °C and 180 rpm until the optical density at 600 nm reached 0.6. High-level expression of EcPepQ in E.coli cells using a T5 promoter-based expression system in 100 mL of antibiotic-containing LB medium was initiated by adding IPTG to a final concentration of 100 μM. After incubation at 25 °C for 12 h, the culture cells were harvested by centrifugation, and were resuspended in 12.5 mL of buffer A (50 mM NaH2PO4, 300 mM NaCl, and 10 mM imidazole) and disrupted by sonication. The cell-free extract was subsequently applied to Ni-NTA agarose (Qiagen, Hilden, Germany) in a column, which had been equilibrated by buffer A. Afterward, the column was washed once with 30 mL of buffer A and eluted with buffer B (50 mM NaH2PO4, 300 mM NaCl, and 250 mM imidazole). Fractions enriched in protein and prolidase activity were pooled for gel electrophoresis and activity assay. Protein concentration was determined using the Bio-Rad protein assay kit with bovine serum albumin as the standard.

Gel electrophoresis and size-exclusion chromatography

Sodium dodecyl sulfate–polyacrylamide gel electrophoresis (SDS–PAGE) was performed on a 12% polyacrylamide separating gel and a 5% stacking gel. For native gel electrophoresis, the experiment was carried out in a vertical slab gel apparatus with 5% stacking gel and 10% separating gel concentration. Coomassie blue R-250 dye was used to visualize protein bands on the polyacrylamide gels.

The apparent molecular mass of the native form of EcPepQ was determined by gel filtration on a Superdex 200 HiLoad 16/60 column (GE Healthcare, Chicago, IL, USA) with 25 mM Tris–HCl buffer (pH 8.0) containing 10 mM Mg(OAc)2 and two mM DTT. A calibration curve was simultaneously established with four reference proteins, including thyroglobulin (670 kDa), aldolase (158 kDa), ovalbumin (43 kDa), and ribonuclease A (13.7 kDa). The elution volumes of reference proteins and the native enzyme were individually recorded to calculate their respective Kav values through the following equation: Kav=(Ve–Vo)/(Vt–Vo)

Where Kav, Ve, Vo, and Vt donate the distribution coefficient of a particular protein, the elution volume of a particular protein, the column void volume, and the total bed volume, respectively.

Activity assays

Prolidase activity was measured by determining the release of proline from Ala-Pro by the recombinant enzyme. The amount of proline in the reaction mixture was estimated by the colorimetric method reported elsewhere (Ghosh et al., 1998). At the beginning of the enzyme reaction, EcPepQ together with 25 mM Tris–HCl buffer (pH 8.0) and other reaction components (one mM Ala-Pro and 0.1 mM Mn2+) were warmed up separately at 60 °C for a short period of time (∼2 min). The catalytic action was initiated by adding 0.1 mL of a suitable dilution of enzyme to the reaction components and sufficient distilled water to bring the final volume up to 0.5 mL. The reaction mixture was incubated at 60 °C for 10 min and then stopped the catalytic action by adding 0.5 mL of acetic acid (3.5 N) and 0.5 mL of ninhydrin reagent (3%, w/v; 3.0 g ninhydrin in a mixture of 60 mL of glacial acid and 40 mL of phosphoric acid). Afterward, the resultant solution was heated at 100 °C for 10 min. The amount of substrate hydrolyzed was calculated from the increase in absorbance at 515 nm. One unit of EcPepQ activity is defined as the amount of enzymes that catalyze the release of one μmole of proline from Ala-Pro per minute at 60 °C and pH 8.0.

Effect of different temperatures (4–80 °C) on the prolidase activity was evaluated with 25 mM Tris–HCl buffer (pH 8.0). The thermal stability of EcPepQ was studied at pH 8.0 and in the temperature range of 4–80 °C for 10 min, and the residual activity was immediately measured under the standard assay conditions. The pH optimum of EcPepQ was determined at 60 °C using 25 mM phosphate-citrate buffer (pH 2.8–5.0), 25 mM sodium acetate buffer (pH 3.6–5.9), 25 mM sodium citrate buffer (pH 4.0–6.0), 25 mM sodium dihydrogen phosphate buffer (pH 6.0–8.0), 25 mM Hepes-NaOH buffer (pH 6.8–8.2), 25 mM Tris–HCl buffer (pH 7.0–9.0), and 25 mM glycine-NaOH buffer (pH 8.6–10.6). The pH stability of EcPepQ was evaluated at 30 °C for 30 min by incubating the enzyme sample (10 μg/mL) at different buffer systems, and the residual activity was then measured under the standard assay conditions. Stimulation of EcPepQ by metal cofactor was determined through assaying the enzymatic activity in presence of 1.0 mM metal ions.

Kinetic parameters of EcPepQ for the hydrolysis of Ala-Pro were determined using 45 μg/mL of the purified enzyme. Substrate concentrations ranging from 0.2 to 14.2 mM were applied to the reaction mixture. The assays were performed at 60 °C for 10 min in Eppendorf tubes in a total volume of 0.5 mL. To determine the kinetic constants of EcPepQ, a Lineweaver–Burk plot was created with data points derived from double-reciprocal transformation.

Circular dichroism spectroscopy

The circular dichroism (CD) spectra of enzyme samples were recorded in the far-UV region (190–260 nm) on the JASCO J-815 spectropolarimeter (JASCO Corporation, Tokyo, Japan) with the light path of one mm. Prior to spectral analysis, the desalted enzyme sample at a final concentration of 0.2 mg/mL was individually incubated at the indicated conditions or treated with different types of organic co-solvents (5–70%, v/v) for 30 min. The spectra were recorded at room temperature (except for the experiment of heat-induced denaturation) using an external circulating water bath. Each spectrum was an average of at least 10 scans and the control signals were taken and subtracted from the respective test to minimize the chances of any false signal generation due to the salts, chaotropic agents or solvents. The data were expressed as mean residue molar ellipticity (deg·cm2·/dmol) based on a residue number of 443 and a mean residue weight (MRW) of 50,176 Da. Mean residue molar ellipticity can be calculated as follows: [θ]mrw = MRW × θobs/10 × c × l, where θobs is the observed ellipticity in degree at a given wavelength, c is the protein concentration in mg/mL, and l is the pathlength in cm. The spectra were also quantitatively analyzed by the DICHROWEB server to estimate the content of secondary structure elements (Lobley, Whitmore & Wallace, 2002).

Thermal unfolding of EcPepQ samples (∼0.15 mg/mL) in 25 mM Tris–HCl buffer (pH 8.0) was determined by monitoring the temperature-dependent changes of the molar ellipticity at 222 nm. In this experiment, EcPepQ samples were heated at scan rates of 0.5, 1, and 2 °C/min. The alterations in molar ellipticity at 222 nm were further fitted with a two-state model (Pace, 1990) to acquire the midpoint (Tm) of the unfolding curve. To explore the possible refolding of denatured EcPepQ, the temperature of thermoelectrically controlled cell holder was reduced by 0.5 °C/min and measurements were taken once every minute.

Fluorescence spectroscopy

Fluorescence analysis of EcPepQ samples was performed at 25 °C in a JASCO FP-6500 fluorescence spectrophotometer with an excitation wavelength of 295 nm. All fluorescence spectra were corrected for the contribution of 25 mM Tris–HCl buffer (pH 8.0), chaotropic agent and organic co-solvents. Before fluorescence spectroscopy, the purified EcPepQ was individually mixed with an appropriate amount of buffer and various amounts of guanidine hydrochloride (GdnHCl) or organic co-solvents to produce solutions with a protein concentration of approximately 0.15 mg/mL. Fluorescence spectra were then recorded for these samples after 30 min of equilibration at room temperature. The emission profiles of EcPepQ samples were recorded from 305 to 500 nm at a scanning rate of 240 nm/min. The maximal peak of fluorescence spectra and the changes in fluorescence intensity were brought together to calculate the average emission wavelength (AEW) (λ) according to Eq. (1) (Royer, Mann & Matthews, 1995). (1) 〈λ〉=∑i=λ1λN(Fi⋅λi)∑i=λ1λN(Fi)

in which Fi is the fluorescence intensity at the specific emission wavelength (λi).

To calculate the transition point and ΔGN-U of GdnHCl-treated EcPepQ, the unfolding data were further fitted with Eq. (2) (Pace, 1990). (2) yobs=yN+yU•e−(ΔG(H2O)N→U−mN→U[GdnHCl]RT)1+e−(ΔG(H2O)N→U−mN→U[GdnHCl]RT)

where yobs represents the observed biophysical signal, yN and yU are the calculated signals of the native and unfolded states, respectively, [GdnHCl] is the concentrations of the chaotropic agent, ΔGN-U is the free energy change for the N↔U process, and the mN-U represents the sensitivity to denaturant concentration.

Results

Enzyme expression and purification

In order to overproduce EcPepQ, the PCR-amplified DNA fragment encoding an approximately 57 kDa protein was digested with BamHI and HindIII, and cloned into the expression vector pQE-30 to yield pQE-EcPepQ (Fig. 1A). Such construction allows the expressed EcPepQ bearing 10 additional amino acid residues at its N-terminus, which facilitates the single-step purification of the recombinant protein by metal-affinity chromatography. Initially, enzyme production by E. coli M15 (pQE-EcPepQ) was evaluated at 28 °C in a five mL medium containing essential antibiotics and 100 μM IPTG for a period of 24 h to select an appropriate time for further optimization. With this information, we then investigated effects of temperature and inducer concentrations on the production of active enzyme by E. coli M15 (pQE-EcPepQ). In these experiments, one mL of bacterial culture was harvested and disrupted by sonication, and the cell-free extract was subsequently analyzed for its specific activity toward Ala-Pro. Results demonstrated that the production of functional EcPepQ achieved a maximum upon a 12 h induction of E. coli M15 (pQE-EcPepQ) with 100 μM IPTG. One major band corresponding to a molecular mass of about 57 kDa was clearly observed in the cell-free extracts of IPTG-induced recombinant cells when the cultivation temperatures were set at above 16 °C (Fig. 1B). It is noteworthy that less amount of the recombinant enzyme were produced by E. coli M15 (pQE-EcPepQ) cultivated at 4 °C. Apparently, the optimal IPTG concentration for the production of active EcPepQ was 100 μM (Fig. 1C). Based on these observations, the best conditions for the high-level production of active EcPepQ by the recombinant cells were the cultivation of E. coli M15 (pQE-EcPepQ) cells at 25 °C, an inducer concentration of 100 μM, and a growth period of 12 h. Under the aforementioned conditions, the specific activity of the cell-free extract from E. coli M15 (pQE-EcPepQ) reached 36.9 U/mg.

Figure 1 Analyses of the soluble proteins and specific activity of E. coli M15 (pQE-EcPepQ) under a specific condition.

(A) Schematic diagram of the key elements of pQE-EcPepQ. (B) Analysis of the crude extracts by SDS–PAGE. Lanes: M, protein size markers; 1, cell growth without IPTG induction; 2, cell growth with 5 μM IPTG induction; 3, cell growth with 10 μM IPTG induction; 4, cell growth with 50 μM IPTG induction; 5, cell growth with 100 μM IPTG induction; 6, cell growth with 500 μM IPTG induction. (C) Effects of incubation temperature and IPTG concentration on the production of active EcPepQ. The amount of active enzyme was determined by measuring the prolidase activity of the soluble extracts shown in (B). These data were a representative of three independent experiments.

The recombinant enzyme in the cell-free extract of E. coli M15 (pQE-EcPepQ) cells was further purified to near homogeneity by an affinity procedure that uses Ni-NTA resin. SDS–PAGE analysis of the pooled fractions exhibited a predominant band with a molecular mass of approximately 57 kDa (Fig. 2A), a finding which is in close agreement with the value deduced from the gene sequence of EcPepQ. The His6-tagged enzyme could be purified nearly16-fold with a yield of 68.4% by single-step affinity chromatography. The native state of EcPepQ was also determined by fast performance liquid chromatography (FPLC) gel filtration (Fig. 2B). The experimental result showed that EcPepQ was eluted from the HiLoad 16/600 Superdex@200 PG column just after aldolase (158 kDa), in a fraction corresponding to a calculated molecular mass of 114.4 kDa (Fig. 2B). Owing to gel electrophoresis analysis of the purified enzyme suggested only one type of subunit (Figs. 2A and 2C), the native state of EcPepQ is most likely to be composed of two identical 57 kDa subunits.

Figure 2 Gel electrophoresis and size-exclusion chromatography of the recombinant enzyme.

(A) SDS–PAGE analysis. Lanes: M, protein size markers; 1, the crude extract of E. coli M15 (pQE-EcPepQ); 2, the enzyme sample after Ni-NTA purification. (B) FPLC analysis. Blue dextran 2000 was used to determine the void volume. The Kav values for the standard proteins and EcPepQ were plotted against the logarithm of their molecular weights to estimate the native molecular mass of EcPepQ. (C) Native PAGE analysis. Lanes: M, protein size markers; 1, the enzyme sample after Ni-NTA purification.

Biochemical characterization of EcPepQ

The effect of temperature on EcPepQ activity was investigated at pH 8.0 over a temperature range from 4 to 80 °C. As shown in Fig. 3A, the maximum activity for EcPepQ was at 60 °C. The pH effect was also studied at 60 °C in the range of 2.7–11.0 (Fig. 3B). The maximum activity was observed at pH 8.0, whereas the enzyme was sensitive to pH shift with more than 80% of activity attenuation at pH 5.0 and 10.0. The thermostability of EcPepQ was evaluated by incubating the enzyme at temperatures between 4 °C and 80 °C for 10 min. As shown in Fig. 3C, the enzyme displayed remarkable stability with more than 90% of the initial activity preserved at temperatures below 30 °C. However, there was a considerable decrease in its activity at temperatures greater than 40 °C. In addition, the effect of pH on the stability of EcPepQ was investigated by incubating the enzyme at 30 °C and different pH’s for 1 h. The experimental results showed that it was stable in the pH value of 8.0, with 65–80% of the full activity at a pH range of 7.0–9.0 (Fig. 3D). However, the enzyme was almost inactive at pH values beyond 5.0 and 10.0.

Figure 3 Effects of temperature and pH on activity (A and B) and stability (C and D) of EcPepQ.

Enzyme assay was performed as aforementioned procedures with one mM Ala-Pro as the substrate. 100% relative activity refers to the percentage of the initial reaction rate obtained by the enzyme at pH 8.0 and 60 °C. The residual activity was expressed as a percentage of specific activity with the untreated sample being defined as 100%. The data are expressed as mean ± SD of three independent experiments.

The effects of several metal ions on EcPepQ activity at a final concentration of one mM are shown in Fig. 4A. It is noteworthy that the addition of Mn2+ ion into the reaction mixture greatly stimulated EcPepQ activity, increasing it by approximately 43-fold as compared with the control. To verify the optimal concentration of Mn2+ ion, EcPepQ activity was assayed under different concentrations of this metal ion. The results clearly indicated that Mn2+ ion at a final concentration of 0.1 mM had the greatest stimulation effect on the prolidase activity (Fig. 4B). However, the presence of Mg2+ and Fe2+ ions in the reaction mixture appeared to slightly enhance the enzymatic activity. Partial inhibition (∼1% inhibition) was observed in the presence of Zn2+ ion, and the strongest inhibitory effect was found in Cu2+ and Ni2+ ions.

Figure 4 Effects of divalent metals (A) and different concentrations of Mn2+ ion (B) on the catalytic activity of EcPepQ.

In these experiments, the sample without extrinsic metal ions was used as a control and the EDTA-treated enzyme served as a negative reference. The data were expressed as mean ± SD of three independent experiments.

The Michaelis–Menten kinetic constants, Km and Vmax, for the purified EcPepQ were determined by using varying concentrations of Ala-Pro. Prolidase activity was measured under standard assay conditions as described earlier and the obtained results were plotted as a graph of enzyme activity (U/mL) against concentration of substrate [μM], which yields a hyperbolic curve with Km and Vmax values. From the graph, Km and Vmax values of EcPepQ were determined to be 8.8 ± 1.1 mM and 434.8 μM/min, respectively. A kcat value of 926.5 ± 2.0 s−1 was further obtained through the Lineweaver–Burk plot.

Spectroscopic studies

Figure 5 presents the far-UV CD spectra of EcPepQ samples. The CD spectrum of the recombinant enzyme displays two strong peaks of negative ellipticity at 208 and 222 nm, indicative of a substantial α-helical content with lesser amounts of β-sheet and random coil (54% α-helix, 9% β-sheet and 21% random coil). The representative peaks for α-helix were significantly diminished at temperatures above 60 °C (Fig. 5A). The thermal unfolding of EcPepQ was initiated at 50 °C and the CD signal at 222 nm was completely disappeared at the denaturation temperature of 75 °C (Fig. 5B). Also, it is evident that the thermal denaturation of EcPepQ followed a two-state process with a well-defined Tm value of 64.2 ± 0.2 °C. To further explore whether the unfolding process was reversible or not, thermal denaturation of the enzyme sample was determined by monitoring the ellipticity at 222 nm at three constant heating rates. After the thermal denaturation went to completion, the enzyme samples were reversely cooled down to 20 °C using the same scan speed. Figure 5B shows the transition curves obtained with EcPepQ solutions at heating rates of 0.5, 1, and 2 °C/min. It is clear that there were no significant differences between these heating rates with respect to the transition temperatures. Besides, it can be seen that the native secondary structure of the enzyme was not recovered immediately after the unfolded protein was cooled down from 90 to 20 °C (Fig. 5B). This observation indicates that the thermal denaturation of EcPepQ is highly irreversible even at the very early stages of the unfolding process.

Figure 5 Far-UV CD and intrinsic tryptophan fluorescence spectra of EcPepQ.

(A) Temperature-dependent CD spectra of the enzyme. Far-UV CD spectra of the enzyme were recorded at the indicated temperatures over the wavelength range of 190–260 nm. (B) Transition and cooling curves of the enzyme . The temperature-induced unfolding of the enzyme was monitored at three different heating rates as aforementioned procedures. The blue line represents the cooling curve of the unfolded protein, which had been heated with a scan rate of 2.0 °C/min. (C) Temperature-dependent fluorescence spectra of the enzyme. Fluorescence spectra of the enzyme were recorded at the indicated temperatures over the emission wavelength range of 305–500 nm.

To probe the change in the tertiary structure of EcPepQ as a function of temperature, the enzyme conformations were also analyzed by fluorescence spectroscopy. Figure 5C shows the intrinsic fluorescence spectra of EcPepQ at different temperatures. Clearly, the fluorescence intensity of EcPepQ was decreased by more than 16% when the enzyme samples were analyzed at temperatures above 60 °C. The tryptophan emission fluorescence spectrum for EcPepQ was maximized at a wavelength of around 333.6 nm. It could be seen that the λmax was individually displayed 1.2, 3.8, and 4.2 nm blue shift when the temperatures were set at 60, 70, and 80 °C (Fig. 5C). Together with the aforementioned CD data, it can be concluded that a significant change in the EcPepQ structure has occurred as a consequence of temperature elevation.

GdnHCl-induced denaturation of EcPepQ

The function of proteins depends on their ability to acquire a unique three-dimensional structure. It is generally known that GdnHCl acts as a classical denaturant to bring about unfolding of proteins by disrupting intramolecular interactions mediated by non-covalent forces. As shown in Fig. 6A, EcPepQ treated with GdnHCl at concentrations of less than 0.5 M retained >80% of the prolidase activity. An increase in concentration up to 1.0 M resulted in 7% of the activity remaining, whereas the enzyme was completely inactivated after treatment with 1.2 M GdnHCl. These results suggest that GdnHCl concentrations below 0.5 M only cause a relatively small change in the molecular structure of EcPepQ and its catalytic activity can tend to be recovered upon removal of the denaturant. Giving the fact that EcPepQ contains a total of seventeen tryptophanyl residues, fluorescence spectroscopy is very suitable for its conformational study. Unfolding of this enzyme at different concentrations of GdnHCl was accordingly performed and the obtained data were shown in Fig. 6B. The AEW that reported on the changes in both fluorescence wavelength and fluorescence intensity was used to calculate the thermodynamic parameter of the unfolding process. As shown in Fig. 6B, EcPepQ started to unfold at 1.2 M denaturant and exhibited a [GdnHCl]0.5,N-U value of 1.98 M, which corresponds to a free energy change (ΔGN-U) of 6.18 kcal/mol.

Figure 6 Concentration effect of GdnHCl on the catalytic activity of EcPepQ (A) and the corresponding changes in the tertiary structure as monitored by AEW value (B).

The purified enzyme at a final concentration of 0.15 mg/mL was incubated with different concentrations of GdnHCl at 30 °C for 10 min. Then, the sample solutions were subjected to measurement of prolidase activity under the standard assay conditions and fluorescence analysis.

Effects of co-solvents on the catalytic activity and molecular structure of EcPepQ

Concentration effects of 12 different organic co-solvents on the enzymatic activity of EcPepQ were investigated with the widely used substrate, Ala-Pro. As shown in Fig. 7, the enzyme exhibited distinct sensitivity to these organic co-solvents. It is noteworthy that the catalytic capability of EcPepQ was well preserved in most of the organic co-solvents tested up to concentrations of 15% (v/v). Isopropanol and tetrahydrofuran (THF) clearly inactivated the enzyme at a concentration of 30% (v/v). Conversely, formamide, methanol, ethylene glycol, and glycerol were found to be the most compatible co-solvents to EcPepQ (Fig. 7). Among the remaining co-solvents, the least level of disability to the functionality of EcPepQ was dimethyl sulfoxide (DMSO), followed by 1,4-dioxane, acetonitrile, acetone and ethanol. It is also important to mention that the enzymatic activity of EcPepQ was primarily decreased as a consequence of an increase in co-solvent concentrations (Fig. 7). Some literatures have shown that DMSO at low concentrations (<10%, v/v) actually promotes the stabilization of some proteins (Huang, Dong & Caughey, 1995; Arakawa, Kita & Timasheff, 2007; Batista et al., 2013). Although the protective mechanism of DMSO remains obscure, protein-solvent preferential interactions might be appropriate to interpret its beneficial effect on the conformational stability of proteins in aqueous solutions (Timasheff, 2002; Arakawa, Kita & Timasheff, 2007).

Figure 7 Effect of different water-miscible organic co-solvents on the enzymatic activity of EcPepQ.

The enzymatic activity was determined in the reaction mixture with different concentrations of organic co-solvents under the standard assay conditions. The residual activity was expressed as a percentage of specific activity with the solvent-free sample being defined as 100%.

To explore the relationship between the functional inactivation and structural disruption of EcPepQ in the water/organic co-solvent mixtures, CD and fluorescence studies were carried out with our laboratory facilities. The far-UV CD spectra of EcPepQ were essentially determined at the co-solvent concentrations that resulted in ≥90% decreases in the prolidase activity, and compared with that measured under solvent-free condition (EcPepQ in 20 mM Tris–HCl buffer, pH 8.0). Considering the presence of DMSO, 1,4-dioxane, and formamide had strong negative interference on the ellipticity signals of EcPepQ, these three co-solvents were excluded from the analysis of CD spectra. As the data presented above, the far-UV CD spectrum of the solvent-free enzyme sample displayed two strong peaks of negative ellipticity at 208 and 222 nm (Fig. 8A). The spectrometric characteristic of EcPepQ was significantly diminished in the presence of water-miscible organic co-solvents, especially in the presence of 70% acetonitrile, 70% methanol, and 60% ethanol. These data clearly indicates that high concentrations of some organic co-solvents lead to profound alterations in the far-UV spectra, reflecting a substantial loss of the secondary structure of EcPepQ.

Figure 8 Far-UV CD (A) and intrinsic tryptophan fluorescence (B) spectra of EcPepQ in the presence of water-miscible organic co-solvents.

Spectral analyses of the enzyme were carried out at 25 °C in either 25 mM Tris–HCl buffer (pH 8.0) or the buffer supplemented with various organic co-solvents at concentrations that led to reductions in the catalytic activity of ≥90%.

Fluorescence emission spectra were also determined upon the excitation of the enzyme samples at 295 nm to monitor changes to the microenvironments of tyrosine and tryptophan residues (Royer, 2006). The fluorescence spectrum of solvent-free enzyme sample exhibited an emission maxima at 330.8 nm (Fig. 8B). Organic co-solvent concentrations that impaired the prolidase activity of EcPepQ to less than 10% generally resulted in either a red shift or a blue shift in the maximum wavelength and a subtle increase in the emission intensity as well. Based on these facts, we speculate that at least some of the fluorescent residues of EcPepQ undergo a relevant change in their local environments (Royer, 2006). Such results further enhance the idea that significant changes in the molecular structure of EcPepQ have occurred upon treatment with high concentrations of organic co-solvents, in good agreement with the experimental results obtained from the analysis of CD spectra.

Discussion

In this study, optimization of cultivation conditions for enzyme production by E. coli M15 (pQE-EcPepQ) was achieved at an incubation time of 12 h, a final inducer concentration of 100 μM and an incubation temperature of 28 °C. Some previous reviews have already documented that the cultivation of recombinant E. coli cells at low temperature and the use of ideal inducer concentration might favor the production of functional proteins (Rosano & Ceccarelli, 2014; Kaur, Kumar & Kaur, 2018). Apparently, the ability to overexpress EcPepQ by the recombinant cells and to purify the active enzyme in a large quantity allows for its molecular characterization and the development of a biochemical process for the remediation of OP compounds.

The structural and functional aspects of protein oligomerization have acquired growing importance over the last two decades. Oligomerization is usually essential for proteins to execute their biological functions and thus is a phenomenon crucial in triggering various physiological pathways (Hasimoto & Panchenko, 2010; Liu, 2015). The majority of protein oligomers forms through non-covalent weak associations, which can often lead to the assembly of subunits into metastable dimers or oligomers (Liu, 2015). A search of 452 human enzymes by Australian researchers has shown that most of these enzymes are present in oligomeric forms and only a third exists as monomers (Marianayagam, Sunde & Matthews, 2004). Based on the experimental results of gel electrophoresis and size-exclusion chromatography, it is apparent that EcPepQ exists as a dimeric protein in aqueous solution. The dimeric status is consistent with earlier findings of various microbial prolidases (Ghosh et al., 1998; Jalving et al., 2002; Yang & Tanaka, 2008; Theriot et al., 2010; Weaver et al., 2014; Huang & Tanaka, 2015).

The reported prolidases exhibit metal-dependent activity, requiring two divalent cations such as Mn2+, Co2+, or Zn2+ for maximal activity (Kitchener & Grunden, 2012). Prolidase is a member of the pita-bread enzyme family (Lowther & Matthews, 2002), which contains dinuclear metal clusters coordinated by identical sets of amino acid residues (two Glu, two Asp, and one His). A previous examination of the crystal structure of EcPepQ has demonstrated that the enzyme features five conservative metal binding residues (Asp246, Asp257, His339, Glu384, and Glu423) to chelate two metal ions (Weaver et al., 2014). Consistently, EcPepQ required Mn2+ ion for maximum activity (Fig. 4). Stimulation by Mn2+ ion has also been observed in other prolidases from both prokaryotic and eukaryotic organisms (Jalving et al., 2002; Lupi et al., 2006; Vyas et al., 2010; Theriot et al., 2010; Huang & Tanaka, 2015).

Microbial enzymes have gained a lot of interest for their widespread uses in a vast array of industries (Singh et al., 2016). However, the stability of enzymes is always a key challenge on the implementation of the biocatalysts in industrial processes, which are designed to operate under extreme temperatures (Silva et al., 2018). Thus, it is imperative to understand as many as possible about how an enzyme loses stability and to what extent we can more precisely control its ideal temperature for catalysis. Interestingly enough, the recombinant prolidase from E. coli NovaBlue, a mesophilic bacterium, had a temperature optimum of 60 °C and a well-defined unfolding transition of 64.2 °C (Figs. 3 and 5). This would enable us to perform EcPepQ-mediated catalysis at high temperatures. Advantages for a biocatalytic process operated at high temperatures have been reported in the literature (Iyer & Ananthanarayan, 2008).

Guanidine hydrochloride is a chemical denaturant that has been widely used to denature proteins and characterize the conformation, stability, and folding/unfolding pathway and mechanism of proteins (Povarova, Kuznetsova & Turoverov, 2010; England & Haran, 2011). Therefore, we also studied the effect of GdnHCl on the enzymatic activity and unfolding of EcPepQ. The activity assay and inactivation kinetics suggest the GdnHCl-induced inactivation of EcPepQ is a monophasic and concentration-dependent process (Fig. 6). The increase in AEW observed up to 1.2M GdnHCl may be caused by tertiary structural rearrangement involving aromatic residues or because of the increased mobility of the local environment of the aromatic residues. Normally, exposed aromatic residues in the unfolded proteins show emission maxima between 348 and 356 nm (Lackowicz, 2006). Treatment of EcPepQ with higher concentrations of GdnHCl has resulted in exposure of buried residues of the native enzyme to the solvent as a red shift in AEW (from 345 to 356 nm).

Effect of the tested organic co-solvents on the prolidase activity of EcPepQ was relatively complex and difficult to precisely elucidate at this stage. However, several key factors, including alterations in conformation and flexibility of enzymes, (de)solvation of active sites, energetics of substrate desolvation, steric hindrance that restricts the accessibility of substrate, and competitive inhibition by co-solvent molecules, are probably responsible for the inactivation of enzymes in the presence of water-miscible organic co-solvents (Kim, Clark & Dordick, 2000; Klibanov, 2001; Graber et al., 2007). In spite of the fact that most of the employed organic co-solvents are strong denaturants, EcPepQ was still clearly active in several water/organic co-solvent mixtures at high concentrations (Fig. 7). This intrinsic capability has also been discovered in a variety of enzymes, including glyceraldehyde-3-phosphate dehydrogenase (Wiggers et al., 2007), carboxylesterase (Mandrich et al., 2012), lipase (Park et al., 2005), NADH oxidase (Toth et al., 2010), and hydrogenase (Serebryakova, Zorin & Karyakin, 2009).

As shown in Fig. 8, organic co-solvent concentrations that caused a dramatic loss of the prolidase activity of EcPepQ was generally detrimental to the molecular structure of the enzyme. The denaturation of proteins by organic co-solvents can be referred to the disruption of the hydration shell around the biomacromolecules or the distortion of the hydrophobic interactions that help to keep the correct folding of proteins (Serdakowski & Dordick, 2008). Thus, a conformational change in the molecular structure may be the primary reason for the inactivation of EcPepQ under higher concentrations of acetonitrile, methanol, and ethanol. This is closely linked with other previous investigations that show a pretty strong relationship between the loss of catalytic activity and the extent of structural integrity of enzymes in water/organic co-solvents mixtures (Tsuzuki, Ue & Nagao, 2003; Secundo et al., 2011). It is also noteworthy that the protein structure of EcPepQ did not alter profoundly in the presence of 50% glycerol, 60% ethylene glycol, 15% THF, and 30% isopropanol (Fig. 8). Several previous reports have made great strides toward causes of activity loss in organic co-solvents (Klibanov, 1997; Kim, Clark & Dordick, 2000; Klibanov, 2001; Graber et al., 2007). Based on these elucidations, other factors instead of structural changes can possibly contribute to the great loss of EcPepQ activity in glycerol, ethylene glycol, THF, and isopropanol.

Conclusion

In summary, a His6-tagged prolidase from E. coli NovaBlue was overexpressed, purified to an electrophoretic purity of ∼96%, and characterized at a molecular level. Similar to most organophosphate-degrading prolidases, the dimeric enzyme was very active in the presence of Mn2+ ion. However, it should be emphasized that a clear change in the protein conformation of EcPepQ was observed upon heat and GdnHCl treatments. These insights into structure–function relationships can guide the protein engineering of EcPepQ to enable the production of more stable variants. Furthermore, a rather unexpected outcome is the fairly high tolerance of EcPepQ toward the tested organic co-solvents. This information is definitely valuable for potential future applications of EcPepQ, particularly for performing its biocatalysis in water/organic co-solvent systems.

Supplemental Information

Supplemental Information 1 Raw data for the specific activity of E. coli M15 (pQE-EcPepQ) under a specific condition

Click here for additional data file.

Supplemental Information 2 Analysis of the cell-free extracts of E .coli (pQE-EcPepQ) by SDS-PAGE

Click here for additional data file.

Supplemental Information 3 Analysis of the soluble proteins of E. coli M15 (pQE-EcPepQ) by SDS-PAGE

Click here for additional data file.

Supplemental Information 4 Raw data for FPLS analysis

Click here for additional data file.

Supplemental Information 5 Raw data for analysis of the native enzyme by non-denaturing PAGE

Click here for additional data file.

Supplemental Information 6 Raw data for effect of temperature on activity and stability of EcPepQ

Click here for additional data file.

Supplemental Information 7 Raw data for effect of pH on activity and stability of EcPepQ

Click here for additional data file.

Supplemental Information 8 Raw data for effect of divalent metal ions on the catalytic activity of EcPepQ

Click here for additional data file.

Supplemental Information 9 Effect of different concentrations of Mn2+ ion on the catalytic activity of EcPepQ

Click here for additional data file.

Supplemental Information 10 Raw data for transition and cooling curves of the enzyme

Click here for additional data file.

Supplemental Information 11 Raw data for far-UV CD and intrinsic tryptophan fluorescence spectra of EcPepQ

Click here for additional data file.

Supplemental Information 12 Raw data for concentration effect of GdnHCl on the catalytic activity of EcPepQ

Click here for additional data file.

Supplemental Information 13 Raw data for concentration effect of GdnHCl on the corresponding changes in the tertiary structure as monitored by AEW value

Click here for additional data file.

Supplemental Information 14 Raw data for effect of different water-miscible organic co-solvents on the enzynatic activity of EcPepQ

Click here for additional data file.

Supplemental Information 15 Raw data for far-UV CD and intrinsic tryptophan fluorescence spectra of EcPepQ in the presence of water-miscible organic co-solvents

Click here for additional data file.

Additional Information and Declarations

Competing Interests

Author Contributions

Data Availability

The authors declare that they have no competing interests.

Tzu-Fan Wang performed the experiments, analyzed the data, prepared figures and/or tables.

Meng-Chun Chi performed the experiments, analyzed the data, prepared figures and/or tables.

Kuan-Ling Lai performed the experiments, analyzed the data, prepared figures and/or tables.

Min-Guan Lin analyzed the data, contributed reagents/materials/analysis tools.

Yi-Yu Chen contributed reagents/materials/analysis tools, prepared figures and/or tables.

Huei-Fen Lo conceived and designed the experiments, authored or reviewed drafts of the paper, approved the final draft.

Long-Liu Lin conceived and designed the experiments, authored or reviewed drafts of the paper, approved the final draft.

The following information was supplied regarding data availability:

The raw data are provided in the Supplemental files.

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
