# Peer review of "High-level expression and molecular characterization of a recombinant prolidase from Escherichia coli NovaBlue"

_PeerJ, doi:10.7717/peerj.5863_

## Round 0.1 · original submission · Minor Revisions

Please address all the critical issues raised by both reviewers and revise your manuscript accordingly

Reviewer 1 ·

Basic reporting

The authors used clear language to describe their design rationales. Figures are well labeled and described, the main text is well referenced.

Experimental design

Methods are described in detail.

Validity of the findings

Findings are valid with robust data presentation.

Additional comments

The article describes a protocol to purify prodilase from E.coli. After purification, the authors employed various biochemical and biophysical techniques to interrogate the properties of the purified prodilase, suggesting promising progress toward the rational adjustment of reaction conditions for prodilase -catalyzed reactions.

The authors used clear language to describe their design rationales. Figures are well labeled and described, the main text is well referenced. Methods are described in detail and the findings are valid with robust data presentation.

This is a nice biophysical study on a protein of interest, and I support its publication if the following minor issues could be adjusted:

1) The abstract needs to be significantly changed. Generally speaking the authors should start with a problem and state rationales why it is important. The current version of abstract simply stated what the authors did, without stating the significance of their work.
2) In the caption for Fig. 2A, the authors should give more information regarding “purified”. Was it after Ni-NTA or after gel filtration? Same goes to 2C.
3) More information should be given to the figure captions in Figs. 3 through 8. In general readers should be able to understand an article just by studying the figures and reading their captions.
4) In lines 304-305, the authors state that “It is clear that the transition point at a heating rate of 0.5 C/min occurred at lower temperature than those of 1 and 2 C/min”. It is however not clear why the authors made this argument; the data appears to be too noisy to draw this conclusion. Please comment.
5) In line 309, “down to 20C”, I suppose the temperature was dropped from 80C? The authors should make the starting temperature clearer.
6) A calculationpeople typically perform from GdnHCl unfolding experimental data is the delta_G of folding of a protein. If the authors could make this calculation it will give readers a direct visualization on the stability of this prodilase compared to other proteins.
7) Axis labels are too small to see in Fig 7. Please modify this figure to make it more readable.

Reviewer 2 ·

Basic reporting

no comment

Experimental design

no comment

Validity of the findings

no comment

Additional comments

This manuscript by Wang et al. demonstrated the overexpression and molecular characterization of a recombinant prolidase from E. coli NovaBlue (EcPepQ). The authors also determined the biophysical and biochemical properties of EcPepQ, which is interesting and informative. However, I recommend the authors to address the following issues to improve the quality of manuscript.

1. The authors claimed that the His-tagged enzyme could be purified nearly 3.0-fold with a yield of 68.4% by single-step affinity purification (lines 255-256). Please describe how the authors calculated the values ‘3.0-fold’ and ‘68.4%’.

2. For the clarity of the figure 2B, it would be better to mark the molecular weights of four reference proteins instead of the numbers (e.g., ‘670 kDa’ instead of ‘1’, ‘158 kDa’ instead of ‘2’, etc.).

3. In figure 3, effects of temperature and pH on activity and stability of EcPepQ were presented. However, the related sentences were a little confused as follows; ‘the maximum activity for EcPepQ was at 60 C (line 265)’, ‘The recombinant enzyme was fully active at temperatures below 40 C (lines 268-269).’ Please explain the difference between relative activity (fig. 3a, b) and residual activity (fig. 3c, d). In addition, for the clarity of the manuscript, I recommend that those paragraphs should be polished up.

4. In figure 7, please indicate the unit of x axis.

5. The legends of figures 3-8 are too simple to fully understand the figures. I recommend that the authors should describe the data in more detail.

6. In figure 8, the authors measured CD and fluorescence spectra of EcPepQ in the presence of various co-solvents with concentrations resulting in >90% decreases in the prolidase activity. Interestingly, although all co-solvents decrease the prolidase activity of EcPepQ with similar levels, several solvents, including acetonitrile, methanol, and ethanol, significantly affected the secondary structure of EcPepQ, but other co-solvents (e.g., glycerol, ethylene glycol, THF, isopropanol) little affected the EcPepQ structure. Please discuss the different effects of those co-solvents on the structure of EcPepQ.

7. Typos
(line 100) (Weaver et al., 2004) should be changed to (Weaver et al., 2014).
(line 190) ‘250 nm’ should be changed to ‘260 nm’.
(lines 356, 366) ‘far-UV spectra’ should be changed to ‘far-UV CD spectra’.

(legend of figure 2) ‘B lue Dextran’ should be changed to ‘Blue Dextran’.
(legend of figure 3) ‘onactivity’ should be changed to ‘on activity’.
(legend of figure 4) ‘ofdivalent’ should be changed to ‘of divalent’.
(legend of figure 4) ‘thecatalytic’ should be changed to ‘the catalytic’.
(legend of figure 6) ‘thecorresponding’ should be changed to ‘the corresponding’.
(legend of figure 7) ‘miscibleorganic’ should be changed to ‘miscible organic’.
(legend of figure 8) ‘tryptophanfluorescence’ should be changed to ‘tryptophan fluorescence’.

---

## Round 0.2 · accepted · Accept

All the critical points raised by both reviewers were addressed and the manuscript was revised accordingly. This revised version can be accepted in its present form.

#